# XDN-Based Network Framework Design to Communicate Interaction in Virtual Concerts with Metaverse Platforms

**Sangwon Oh [1], Kwangmoo Chung [1], Ibrahim Aliyu [1], Minsoo Hahn [2], Il-Kwon Jeong [3], Cho-Rong Yu [3], Tai-Won Um [4,*] and Jinsul Kim [1,*]**

1. Department of ICT Convergence System Engineering, Chonnam National University, Gwangju 61186, Republic of Korea; osw0788@gmail.com (S.O.); chungkm1250@gmail.com (K.C.); ibal2010@yahoo.com (I.A.)
2. Department of Computational and Data Science, Astana IT University, Astana 010000, Kazakhstan; m.hahn@astanait.edu.kz
3. Electronics and Telecommunications Research Institute, Daejeon 34129, Republic of Korea; jik@etri.re.kr (I.-K.J.); crryu@etri.re.kr (C.-R.Y.)
4. Graduate School of Data Science, Chonnam National University, Gwangju 61186, Republic of Korea
* Correspondence: stwum@jnu.ac.kr (T.-W.U.); jsworld@jnu.ac.kr (J.K.); Tel.: +82-62-530-1808 (J.K.)

**Abstract:** In the trend of transforming existing systems and services into non-face-to-face models, the concert industry is also showing movements toward transitioning to virtual formats. Physical concerts in the real world require venues that can accommodate hundreds to tens of thousands of spectators, but non-face-to-face methods that can accommodate large audiences face various limitations. Moreover, to elevate the satisfaction level of virtual concert attendees to that of real-world concerts, it is important to implement interaction between performers and audiences. Modern metaverse platforms apply cutting-edge network technologies to accommodate numerous users within a single channel. Many researchers are adopting network technologies such as SDN (software-defined networking) and CDN (content delivery network) to set up a virtual concert that can accommodate large audiences. In this paper, we propose a network framework to be designed for the composition of virtual concerts. In particular, we separate a channel dedicated to interaction in order to provide an immersive experience of exchanging interactions between performers and audiences. As massive audiences transmitting interaction data to the performer in a 1:N format can lead to problems with acceptance and latency, this study introduces a concept of a channel form called 'Zone' and proposes an interaction data channel network framework that does not compromise immersion. The proposed framework supports tasks for effectively transmitting interaction data using network technologies for metaverse platforms such as XDN and clustering algorithms such as fuzzy c-means. We also suggest a CDN-based architecture that can ensure low latency for performers to transmit interaction data to the audience.

**Keywords:** cloud edge; content delivery network; experience delivery network; metaverse; multiuser virtual environments

## 1. Introduction

As global pandemics limit outdoor activities, various arts and cultural services and systems are attempting to operate non-face-to-face. In particular, metaverse platforms, which offer immersive environments to overcome the lack of physical presence, a limitation of online contexts, are garnering significant attention. To provide immersive experiences, 3D metaverse games including Roblox and Minecraft are utilized [1–4]. Recently, with the development of various metaverse games and platforms, many planners and researchers are actively conducting studies and making efforts to implement the metaverse. Likewise, in the concert industry, musicians are starting to host virtual concert events using various metaverse platforms. This trend of major artists participating in virtual concert services

is accelerating [5–7]. For instance, startups providing virtual concert platforms, such as WAVE, where avatars perform live concerts, have attracted investments of up to 30 million dollars. These industries are expanding their content by producing concerts with world-class artists such as Justin Bieber and Ariana Grande, as shown in Figure 1, against the background of enormous capital.

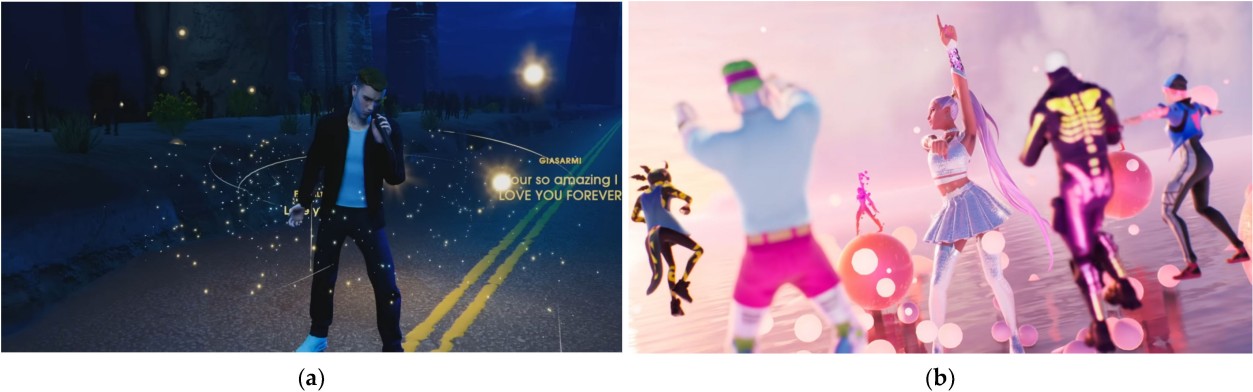

(**a**)  (**b**)

**Figure 1.** (**a**) Virtual concert performed by Justin Bieber [8] and (**b**) Virtual concert performed by Ariana Grande [9].

The metaverse platform industry is researching and integrating various technologies, including volumetric technology, to provide users with immersive experiences [10]. Volumetric technology captures human motion in a green screen studio equipped with hundreds of 4K-level cameras, creating a 360-degree stereoscopic image. Human objects in a virtual space created using volumetric technology are typically referred to as digital humans. While volumetric content has the advantage of enabling a life-like avatar representation of performers compared with traditional virtual concert content, interaction between performers and users has been limited to a few dozen simultaneous connections within the metaverse channel because of real-time concurrent user restrictions. Interaction in concerts is a means for performers and audiences to communicate bidirectionally. Implementing an interactive environment in the virtual concert world of the metaverse platform can enhance QoS for both parties. However, when the audience and performers exchange interaction data during a virtual concert, the amount of traffic that needs to be processed in parallel increases exponentially with the number of audience members.

To construct a metaverse network for virtual concerts, many providers create concert venue worlds and environments using game engines. To efficiently create a metaverse world, world creators use basic network functions provided by the game engine. However, building a virtual concert with a single function results in handling various types of data occurring in the concert venue world within a single network framework. In particular, interaction data exchanged between performers and users require an end-to-end network, potentially escalating the overall network framework's requirements. Moreover, the performer values the visual experience of receiving interactions from the audience more than the specificity of the transmitting party, so it is not necessary to implement 1:N type communication depending on the number of audience members. For instance, if 1000 audience members enter and all 1000 interaction data they send must be processed, a real-time concert can be compromised as a result of exponentially increased network traffic.

Hence, in this paper, we employ bidirectional network channels to segregate the network channel transmitting interactions. This study proposes a framework where the audience in the virtual space is divided into $n$ zones to receive interaction data from a massive audience, with each zone summarizing its interaction data for transmission to the performer. In Section 2, we explain related technologies and the virtual concert network, while Section 3 outlines the CDN-based network framework that we aim to design in this

paper. Section 4 concludes this research, detailing future experimental plans and issues to address.

## 2. Related Research

In this section, we describe the basic approach to constructing the network framework, the architecture of the metaverse network for existing virtual performances, and the background and algorithm of the approach used in this paper to process audience interaction data. Section 2.1, Section 2.2, Section 2.3 introduce the basic technologies of the network structure proposed in this paper, Section 2.4 describes the network architecture required to create a virtual performance on the existing metaverse platform, and Section 2.5 presents an algorithm that can function within the network structure.

### 2.1. SDN (Software-Defined Networking)

Software-defined networking (SDN) is a concept of defining and controlling networks through software, a framework designed and built to manage traffic handling methods and various functions in a centrally controlled system by separating the data and control domains. This concept was devised to overcome the complexity and flexibility limits of traditional hardware-centric network structures and to implement user-centric networks. It started with a paper published in August 2007 by Stanford University, proposing a way to facilitate network management and enhance security [11]. SDN provides an open interface that enables the development of software capable of controlling connectivity provided by a set of network resources and network traffic flow, along with potential inspection and modification of traffic performed in the network. As illustrated in Figure 2 below, SDN is composed of three layers, emphasizing four features: the central separation of the Infrastructure Layer and Control Layer, a central controller, an open interface between the Control Layer devices and Infrastructure Layer devices, and network programming by external applications [12].

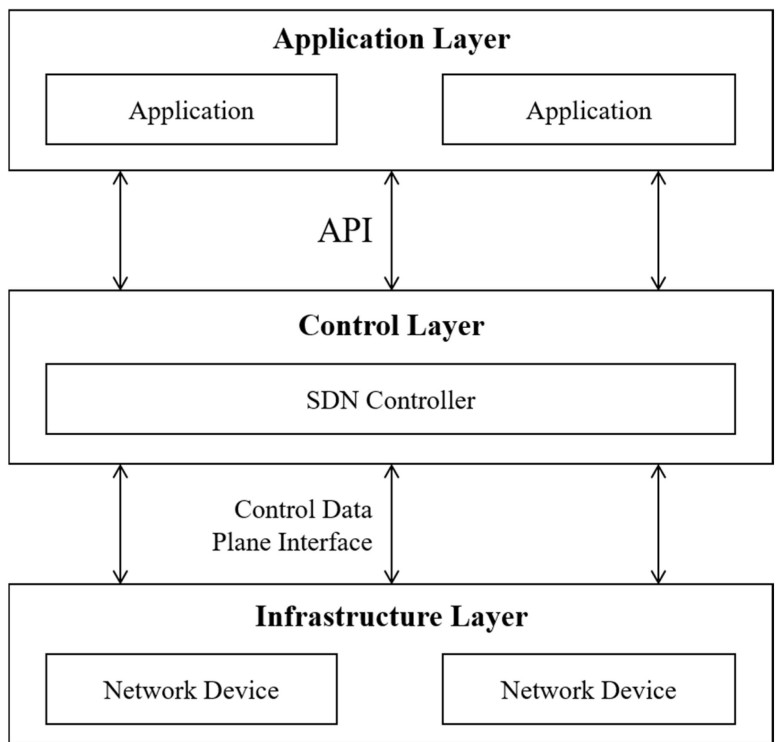

**Figure 2.** Architecture of SDN (software-defined networking) [13].

The Application Layer is where various programs utilizing SDN exist. It enables the use of various functions across the network through the APIs provided by the Control

Layer. These APIs are used for interaction between the Application Layer and the Control Layer and can be used by the Application Layer to receive information related to the Control Layer. The Control Layer is the layer that centrally handles the flow of network data, responsible for determining the routing and forwarding methods for specific packets. It controls the equipment of the data layer through the Control Data Plane Interface and provides an interface abstracting the network's functions to the Application Layer through APIs. The Infrastructure Layer is a layer composed of network devices capable of packet forwarding and processing. It houses the flow table, which is responsible for the actual transmission of data. It operates by receiving routing information from the Control Layer through the Control Data Plane Interface. These layers are controlled through interfaces. Because traditional network devices are single-box, there has been difficulty in changing or manipulating the equipment. Therefore, SDN is designed so that control is possible as long as one knows the standard interface, enabling escape from hardware vendors.

### 2.2. CDN (Contents Delivery Network)

The content delivery network (CDN) was proposed to enhance the traditional content delivery method, which relied on a single large-scale server and could not guarantee quality while providing content to all users via the internet [14–16]. CDN provides multiple servers responsible for numerous users to prevent them from receiving content through unreliable internet bandwidth. Figure 3 below illustrates the servers constituting a CDN. Among the components of a CDN, the Staging Server receives content from the Content Provider and stores content to be distributed to other servers. All content distribution begins at the Staging Server and eventually arrives at the Edge Server, which is the server that delivers content to users and serves as the target for content distribution.

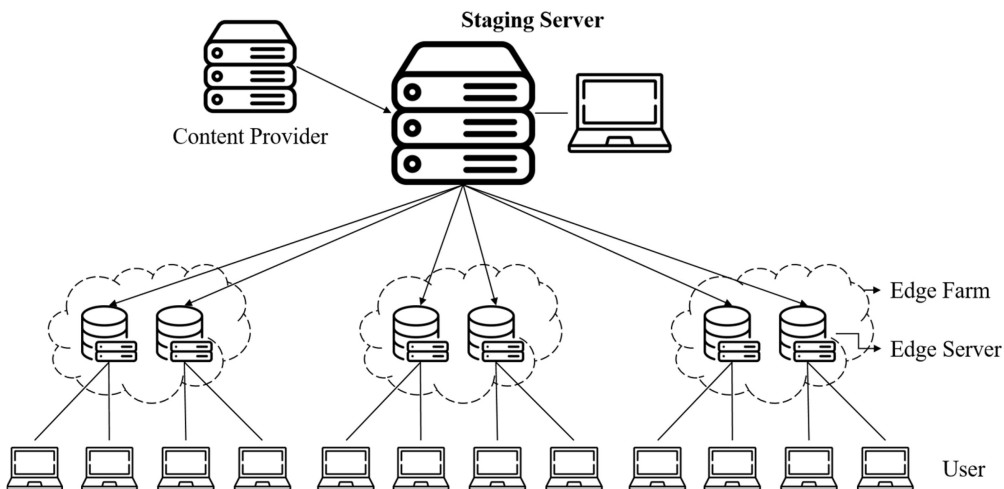

**Figure 3.** Architecture of CDN servers [16].

A typical CDN is a distributed network of servers that can efficiently deliver content to users. It minimizes latency by storing cached content on Edge Servers located near the point-of-presence (POP) of the end user. Through the application of edge computing, it enables proactive responses based on the situational awareness of the end users, depending on the type of service. In metaverse virtual concerts, interactions between the performer and user, as well as between users, are crucial for creating a sense of immersion and realism. Consequently, securing a bidirectional data channel is essential. However, there are limitations with the current CDN structure to facilitate this [16].

### 2.3. XDN (Experience Delivery Network)

Extended reality (XR) is a hyper-realistic technology that creates an expanded world by utilizing the technology of mixed reality (MR), which combines the advantages of virtual

reality (VR) and augmented reality (AR). Services that support extended reality (XR), such as virtual concerts, provide immersive content to users [17,18]. Delivering immersive content in a CDN environment faces the following challenges:

- Expansion to accommodate massive user-generated content is required.
- Interactive experiences, real-time delivery of content composition, and content synchronization are necessary.
- Creation of a 360-degree view; addition of text and image overlays; and real-time collection, processing, and distribution of content via wireless networks are required.

Thus, to address these issues, experience delivery network (XDN), which overcomes the limitations of the conventional CDN, has been recently proposed. XDN, an evolution of CDN, enables immersive content experiences including XR and metaverse through the application of 5G and edge computing [19,20]. In services based on XR, for instance, ultra-high-definition streaming data from a 360-degree view are transmitted, or additional content data are overlaid onto the existing data for transmission, providing users with a highly immersive content experience [21]. Various microservices for collecting, controlling, and distributing media in XDN can be seen in Figure 4 below. The Media Control Function (MCF) provides functions to control authentication, session management, and clustering of virtual concert audiences and Zones, processed at the edge close to content providers and consumers. The Media Distribution Function (MDF) has adopted viewport-dependent streaming based on a tile-based structure to ensure compatibility and scalability with the OMAF format, which supports 360-degree media [22,23]. Using this format and streaming, it handles the distribution of zone-specific media. The Media Ingestion Function (MIF) deals with the ingestion of various types of media depending on the environment of the performer and audience.

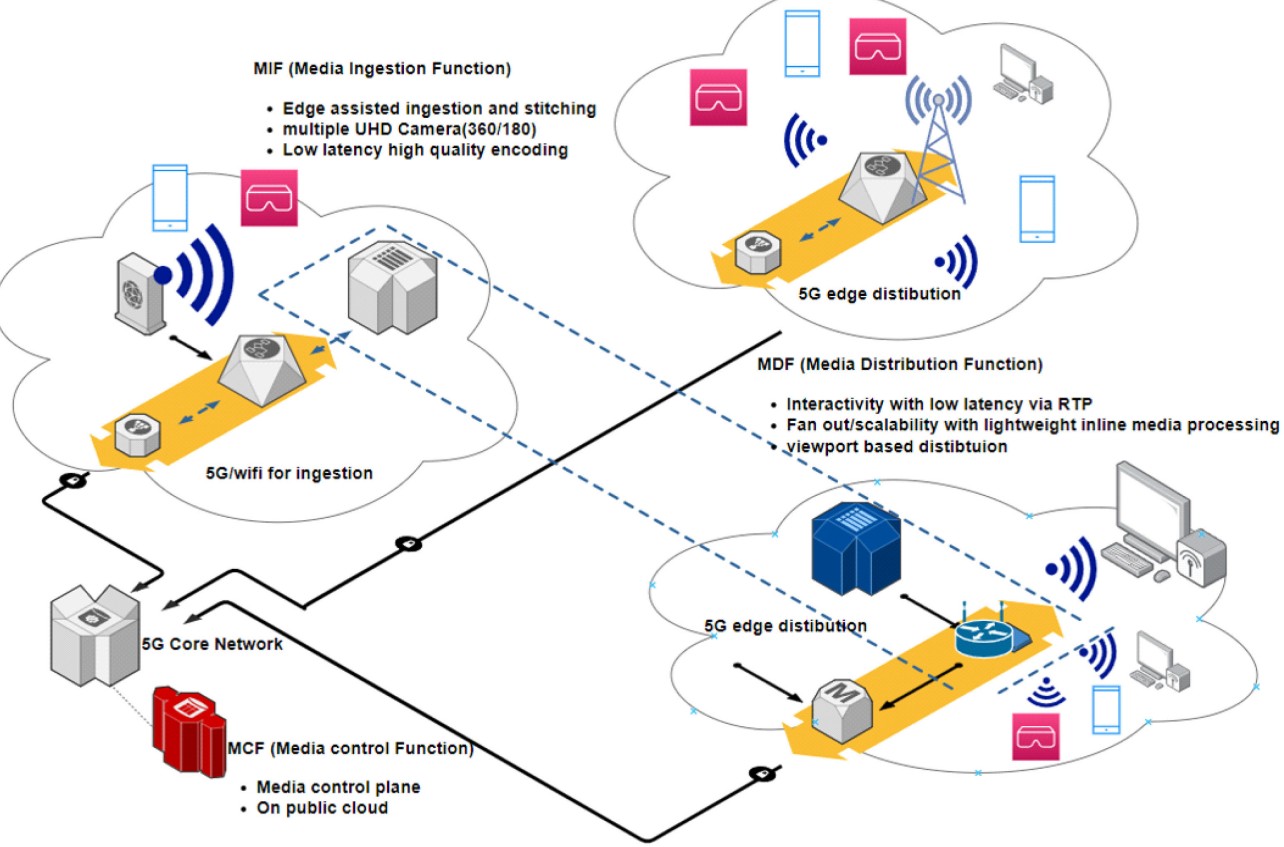

**Figure 4.** Schematic of XDN (experience delivery network).

Figure 5 illustrates the sequence diagrams of microservices typically required when constructing an XDN. These examples represent operations performed in a zone network architecture within a WebXR-based system that supports XDN. Figure 5a represents the Media Ingestion Signaling Flow, illustrating the process of individual users ingesting from the MIF, while Figure 5b represents the Media Distribution Signaling Flow, showing the process of distributing streaming data based on WebRTC from the MDF to each user. Here, WebXR is a framework providing XR experiences based on WebRTC. As shown in Figures 4 and 5, one can adopt microservices of XDN for ingesting or distributing media streaming data and, furthermore, it can be a good choice for transmitting immersive content.

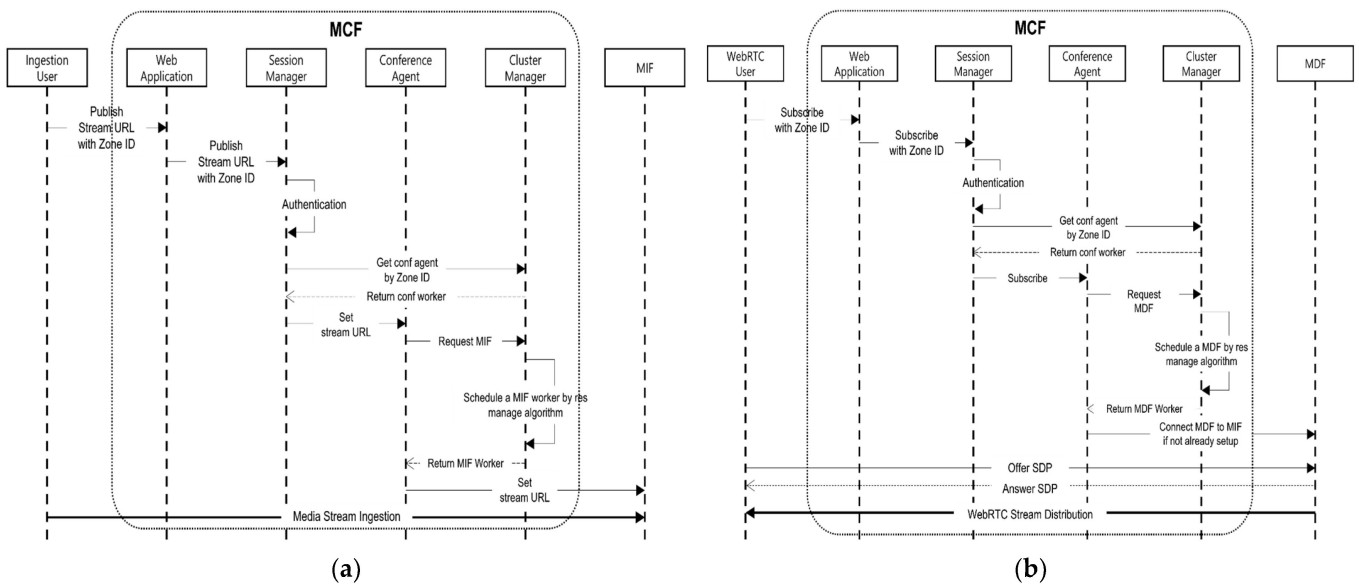

(**a**)  (**b**)

**Figure 5.** (**a**) Sequence diagram of MCF and MIF in the XDN system based on WebXR (WebRTC) and (**b**) sequence diagram of MCF and MDF in the XDN system based on WebXR (WebRTC) [17].

XDN architectures can reduce bandwidth consumption and latency and provide the scalability needed for handling abnormal traffic loads, improving server performance. Therefore, in order to design a real-time adaptive metaverse-based network with ultra-low latency, it is necessary to design a robust network architecture for the XDN [17,24].

### 2.4. Metaverse Network for Virtual Concert

To design the network structure constituting the virtual concert, an analysis of requirements specialized for various content and services must first be conducted. In particular, for the communication of interaction data between performers and the audience, network designers need to perform tasks to derive requirements for interactions and network functions. We assume an environment capable of constituting a virtual concert world where thousands can participate simultaneously, and ensuring free interaction between performers and users within the concert hall. As shown in Table 1, network functions for interactions can be derived.

**Table 1.** Network function for interaction data based on requirements [16].

| Requirements | Network Function |
| --- | --- |
| Pre-stored user motion information (joint variations) Emotional expressions through emoticons | Selection of interactions at each step and definition of data structures are necessary. |
| Text message (chatting) Voice | Joint variation information in 3D vector form is delivered through data streaming. |

Before designing a network for the communication of interaction data, the network for the overall virtual concert must first be designed. The metaverse virtual concert platform allows subscribers to access the platform and participate in their desired virtual concert service channel according to role distinction, managed through subscriber management and concert information management. To provide scalable server operations and variable load balancing features, the virtual concert platform is loaded onto a cloud-based server. This allows for an increase in the number of physical subscriber servers as the number of simultaneous users increases. As shown in Figure 6, pre-defined fixed stage-related assets in the planning process for the virtual concert hall and stage equipment are downloaded before the real-time concert, and during the real-time concert, only real-time data provided from the performers are streamed.

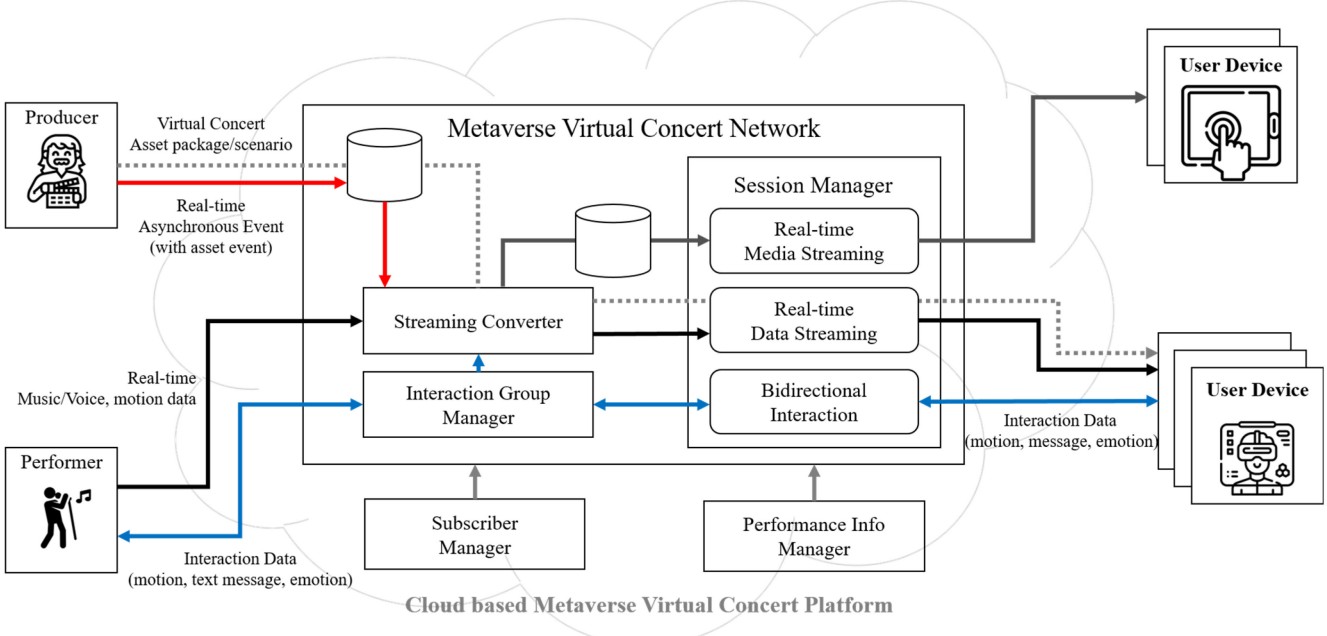

**Figure 6.** Metaverse virtual concert network design [16].

A virtual concert takes the form of users connecting to a virtual channel, and this virtual channel can vary dynamically. The session manager manages and controls the physical channels within the virtual channel, and the producer takes on various roles according to events that occur asynchronously during the concert. The interaction data discussed in this paper are created and delivered bidirectionally between the performer and the audience. The virtual concert network can be configured to selectively provide only real-time media streaming for user devices that have difficulty generating motion data.

### 2.5. Fuzzy C-Means Algorithm

In this paper, we propose a metaverse network architecture for virtual concerts utilizing SDN and XDN, as previously introduced. To efficiently operate a network targeted at a massive multiplayer base, such as a virtual concert or game, clustering users (nodes) according to an appropriate standard is required. Clustering techniques are broadly divided into hierarchical clustering and partitional clustering. Hierarchical clustering uses a hierarchical tree model to perform clustering by integrating individual objects into sequentially and hierarchically similar objects or clusters. Partitional clustering initially sets the number of clusters and then searches for the optimal center of each cluster to perform clustering. In this study, we chose partitional clustering because we need to distribute network resources according to the clustering results.

Partitional clustering is divided into hard clustering when each data must belong to a single cluster and soft clustering when it can belong to multiple clusters. In soft clustering, each data can potentially belong to multiple clusters. The fuzzy c-means algorithm is a representative soft clustering technique. Dunn has devised a method using fuzzy segmentation by expanding the k-means algorithm and presenting the objective function as shown in Equation (1) below [25,26].

$$\sum_{k=1}^{K} \mu_{ik} = 1, \quad 0 < \sum_{i=1}^{n} \mu_{ik} < n \tag{1}$$

$$\min J_m = \sum_{k=1}^{K} \sum_{i=1}^{n} (\mu_{ik})^m d(x_i, \, c_k)^2 \tag{2}$$

As fuzzy c-means clustering is an algorithm that calculates the probability of each data belonging to a certain cluster as a probability, the sum of $\mu_{ik}$, which is the weight of the probability that data $x_i$ belongs to each cluster $c_1$, $c_2$, $\ldots c_k$, is 1. In addition, to ensure the effectiveness of the clustering analysis using fuzzy logic, each cluster $c_k$ has at least one data with a non-zero weight, and all of the weights of the cluster cannot be 1. Therefore, the partitioning condition is satisfied as shown in Equation (1). The clustering algorithm, like any other hard clustering, proceeds by minimizing $d(x_i, c_k)$, which is the distance between the observation and the fuzzy cluster centroid, i.e., the optimal cluster is found by minimizing the $J_m$ objective function, as shown in Equation (2). Here, $m$ is a constant greater than 1 and is the fuzzifier that controls the value of the weights.

$$C_k = \frac{\sum_{i=1}^{n} \mu_{ik}^m x_i}{\sum_{i=1}^{n} \mu_{ik}^m}, \quad k \in [1, K] \tag{3}$$

$$\mu_{ik} = \frac{\left\{ \frac{1}{d(x_i, c_k)^2} \right\}^{\frac{1}{m-1}}}{\sum_{j=1}^{K} \left\{ \frac{1}{d(x_i, c_j)^2} \right\}^{\frac{1}{m-1}}} \tag{4}$$

We need to calculate a partial derivative to estimate each variable that minimizes $J_m$. Equations (3) and (4) are the solutions of the system of equations after calculating the partial derivative with respect to $c_k$ and $w_{ik}$, respectively, and setting their values to zero. As the value of the fuzzifier $m$ increases, the mean of each cluster moves closer to the overall mean, so it becomes increasingly fuzzier to classify the data into one cluster. Generally, $m$ is set to 2 for computational convenience, but in this study, fuzzy c-means clustering is performed through various m values. Because the fuzzy c-means algorithm specifies the number of clusters and weighting coefficients and has a different optimization procedure from the k-means algorithm, it offers superior results in various experiments compared with the k-means algorithm [27–29]. This algorithm has been employed for clustering in a wide range of industries; in this study, we utilized it to cluster audiences in the virtual concert environment [30–32].

## 3. Network Framework Design to Communicate Interaction in Virtual Concerts

The network framework proposed in this paper represents the biggest difference between the network configuration of the existing metaverse-based virtual concert system and the network configuration described in Section 2, Section 3, Section 4. To solve the problem of exponentially increasing the number of users accessing the virtual concert channel as the size of the existing virtual concert users is expanded to a large scale, we applied virtual channelization technology. In an environment where the number of channels changes variably, the bidirectional interaction data channel was separated to ensure that interaction data can be well synchronized and represented in the virtual concert channel.

Figure 7 below proposes a method for sending data, including information to identify the interaction target, for the audience to transmit interaction data to the performer.

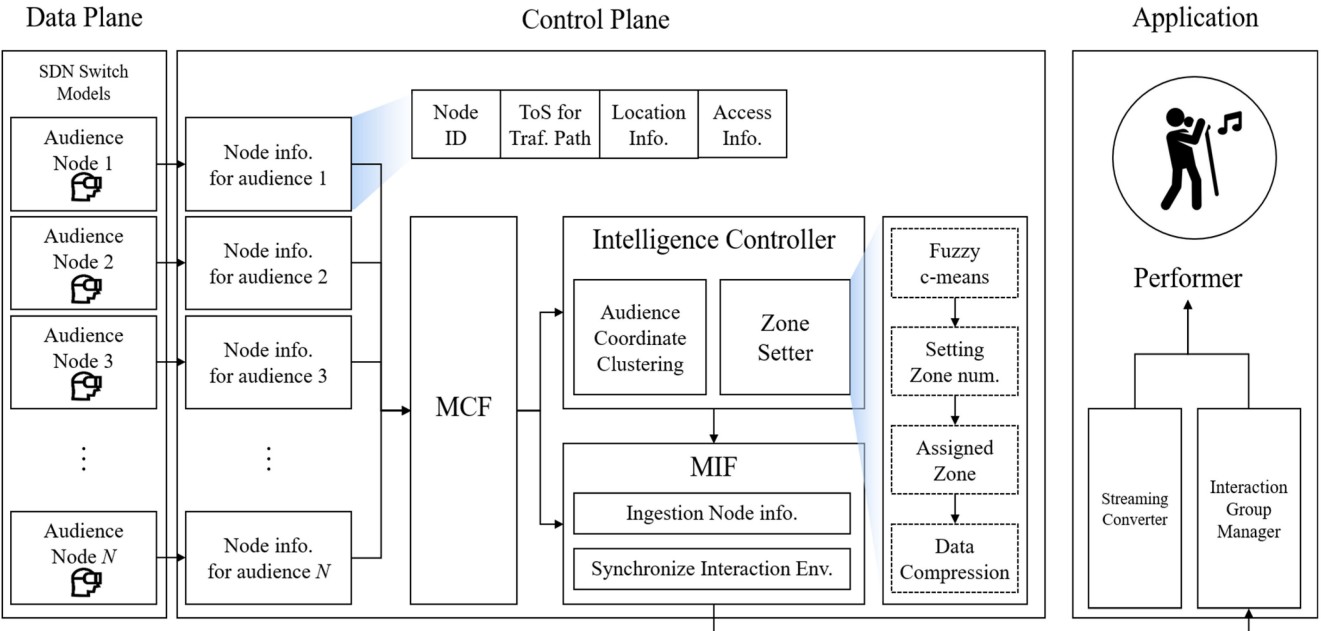

**Figure 7.** Network framework that transmits audience interactions.

Considering the virtual channel and the dynamically changing network environment, we adopted an SDN (software-defined networking) type of framework. The Data Plane is composed of node models that receive information from the audience in the virtual concert and perform SDN switch functions. The node and the process models inside the node send their information to the Control Plane, which is then delivered to the MCF (Media Control Function) microservice of the XDN for the preparation of authenticating audience information and assigning zones. Some studies receive and store the node's information in the form of a pointer using a shared memory interface [33], but this study does not deal with specific packet processing methods for synchronizing other network depths outside of the interaction channel. The main information of the node received from the Control Plane includes the node's Id, ToS for traffic path information, location in the virtual concert, and access environment information. Before the virtual concert begins, if the entire audience has entered, the MCF sends the node information to the Intelligence Controller.

Once the MCF receives information about the audience that has entered the virtual concert, it sends the node's packet information to the Intelligence Controller to create zones and assign audiences to each zone. First, 2D (which could also be 3D depending on the nature of the world) coordinates are generated through the audience location coordinates within the virtual concert world, and clustering is carried out in the Zone Setter. Various metrics including the elbow method are used to determine the appropriate number of clusters c, and the fuzzy c-means algorithm is performed accordingly to generate the number of zones and centroids. Once the coordinates of the centroid within the virtual concert are determined, the Zone Setter performs clustering and assigns zones to each node. Then, the MIF ingests the node information, including zone information, after compressing it.

After the concert starts and interaction data are sent from the node, the process mentioned above is followed until the MIF accumulates interaction data within zones. The MIF classifies the types of interaction data accumulated over a period of time, measures the amount of data transferred for each type, and sends it all to the Interaction Group Manager. This information is then sent entirely to the Interaction Group Manager. The

types of interactions can be configured as shown in Table 2, but the virtual concert producer has the discretion to specify more diverse interaction data types.

**Table 2.** Type of interaction data when used in virtual concerts.

| Kind of Interaction | Details (0 . . . N) |
|---|---|
| Gesture | Jump, Hurray<br>Waving arms |
| Emoticon | Smile, Nyah, Weeping<br>Shoot Heart Emoticon |
| Text (Default Message) | "Hello!"<br>"Nice to meet you."<br>"I love you."<br>"Very good."<br>"Wonderful!" |

The interaction data from the node are sent to the performer and, to avoid affecting the performance provided in the form of streaming, the interaction transmission channel is kept separate. Furthermore, to synchronize the streaming and interaction, the MIF performs the task of synchronizing both channels. If the performer sends an interaction to the audiences, they bypass the separate MIF or Intelligence Controller and directly transmit interaction data to all nodes. As shown in Figure 8, not only does the performer provide performances through the streaming converter as a central server in the form of a CDN, but it also sends interaction data to each Edge Farm, supporting low-latency communication. Performers can configure the following CDN-based edge servers to transmit interaction data to large audiences. Edge servers are critical components in the proposed network structure to reduce latency and to manage traffic efficiently. They are strategically placed closer to the nodes (audience) and the performer, which allows data to travel shorter distances, resulting in faster data delivery. Here is a brief conceptual overview:

- **Central Server (Performer)**: This is where the live performance happens. The performance data are streamed in real time from this location. Interaction data from the performer, if any, are also sent from here.
- **Streaming Converter**: The streaming converter takes the raw performance data (motion, audio, and voice data) from the central server (performer) and converts them into a format suitable for transmission over the network. It might involve compression or other methods to ensure efficient data transmission.
- **Edge Servers (Edge Farm)**: These are servers placed closer to the end users (nodes). They receive the performance stream from the central server via the streaming converter and then relay these data to the end users. Similarly, interaction data from the users are collected and sent back to the central server through these edge servers.
- **Nodes (Audience)**: These are the end users or audience members in the virtual concert. They send interaction data and receive performance data through the edge servers.

This setup ensures that data does not have to travel long distances all the way from the performer to each individual user, and vice versa. Instead, the data travel to nearby edge servers, which can relay them to the users or back to the performer as required, significantly reducing latency and enabling real-time interactions. However, because the above method needs to consider scalability, it is an option worth designing if the virtual performance platform is configured based on the cloud as shown in Figure 6.

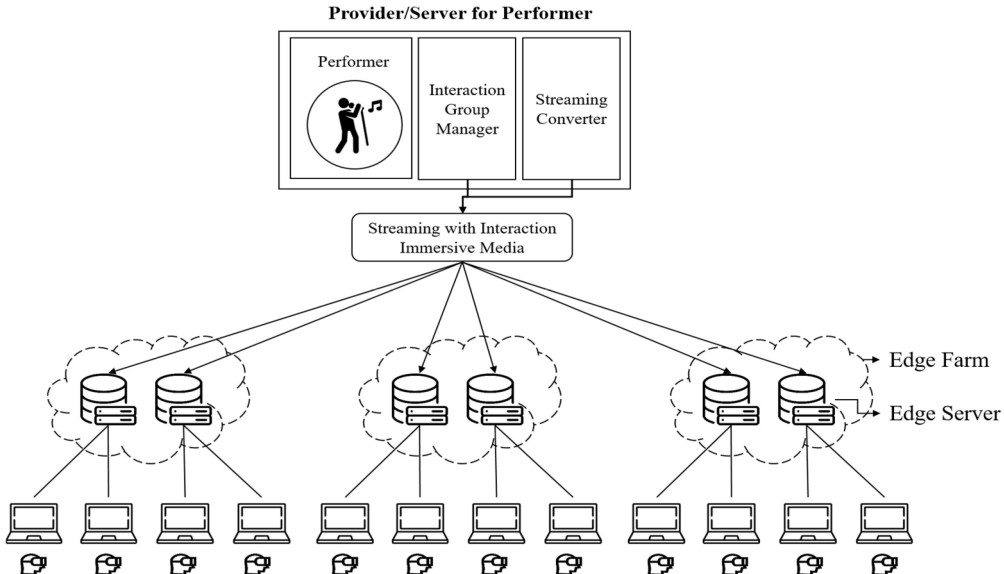

**Figure 8.** CDN-based network architecture for transmitting interaction to large audiences.

## 4. Simulation of Interaction Telecommunication in Virtual Concerts

Simulations were conducted to validate the model for clustering audiences in the Zone Setter, a framework for transmitting audience interaction data. To find the parameters that perform optimal clustering, we generated audiences located within the virtual performance world and performed fuzzy c-means (FCM) clustering based on their location values. The hyperparameters to be considered in the FCM algorithm to cluster the zones are as follows:

- **Audience placement in the virtual concert world**: FCM clustering is an algorithm that updates the probability of belonging to each cluster by generating a vector with a preset number of columns for each piece of data. The performance and execution time of clustering according to the number of audiences has a linear relationship, and the realistic placement of audiences can determine the reliability of the clustering results.
- **Number of zones (number of clusters)**: As this study uses partitioned clustering, the number of clusters must be set in advance to perform clustering. In the actual metaverse environment, the number of zones may vary depending on the specifications of the network equipment and the environment, so the number of server environments that can be allocated varies depending on the performance.
- **Number of clustering iterations (iter)**: As iterations increases, the FCM algorithm is performed once. Initially, a centroid is assigned at the same coordinates or random coordinates, and the centroid is updated by the value of iter. Because clustering in the Zone Setter needs to be fast to provide the audience with a lag-free immersive experience, it is important to quickly capture the iteration that approaches the optimal clustering result.
- **Fuzzifier**: Fuzzifier is a hyperparameter that has a great impact on the performance of FCM, and researchers usually use experimental or heuristic methods to set the fuzzifier [34]. Through simulation, various experiments were conducted in this study to determine the appropriate fuzzifier to port to the Zone Setter.

To recreate the environment where the audience is concentrated around the performer's coordinates in a real performance, we used a hypergeometric distribution to generate random audience placement. The number of success states and the number of failure states were adjusted to be equal to each other to create a distribution of coordinates clustered around a central coordinate, which was set to be the performer's coordinate. For this simulation, we set up a total of four random audience placements for a certain number of audience members, as shown in Figure 9.

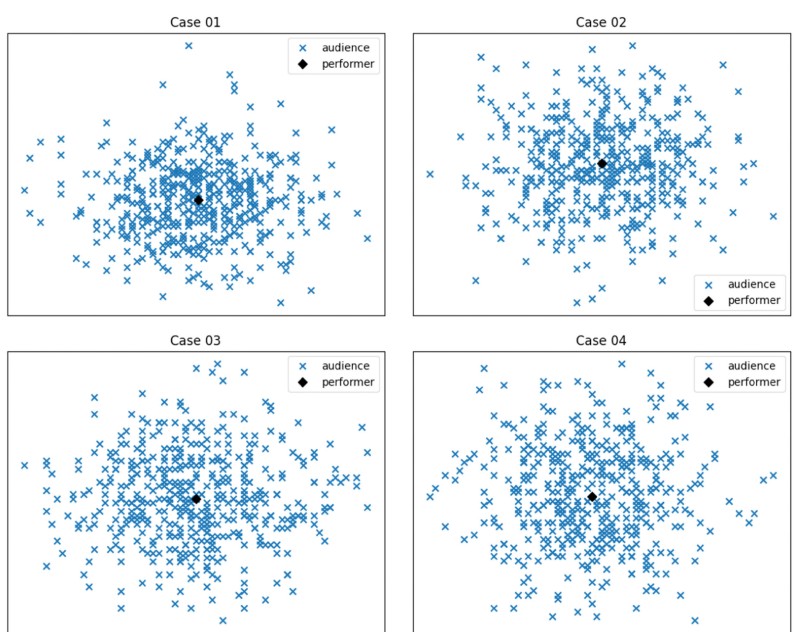

**Figure 9.** Example of random audience placement plot (500 people audience).

We performed FCM clustering on audiences of 500 people to see how the centroid changed with fuzzifier. We used the scenario of case 01 in Figure 9, and the number of clusters was set to 5. The total number of iterations was 50, and the FCM algorithm was performed with 5 different fuzzifier values. The results showed that FCM clustering with a fuzzifier between 1.1 and 1.5 showed a high initial change in each iteration of the algorithm, and then the change converged to 0 after a certain number of iterations, while, with a fuzzifier above 1.7, there was almost no change in the initial centroid coordinates overall. Therefore, it is recommended to set the fuzzifier to at least 1.5 or less to achieve fast clustering results in fewer iterations. We also conducted an experiment to check the variation in the optimal number of clusters depending on the fuzzifier when exploring the optimal number of clusters using the elbow method. As in the previous experiment, we set five fuzzifier values to check the change in inertia (sum of Euclidean distance between data and clusters) from the minimum number of clusters of 2 to the maximum number of clusters of 27. As a result, we found that the change in inertia was quite similar for fuzzifiers from 1.1 to 1.5, and the change in inertia decreased from 1.7. Therefore, it is recommended to set the fuzzifier between 1.1 and 1.5 for consistent optimal cluster number determination. The results of these experiments are shown in Figure 10.

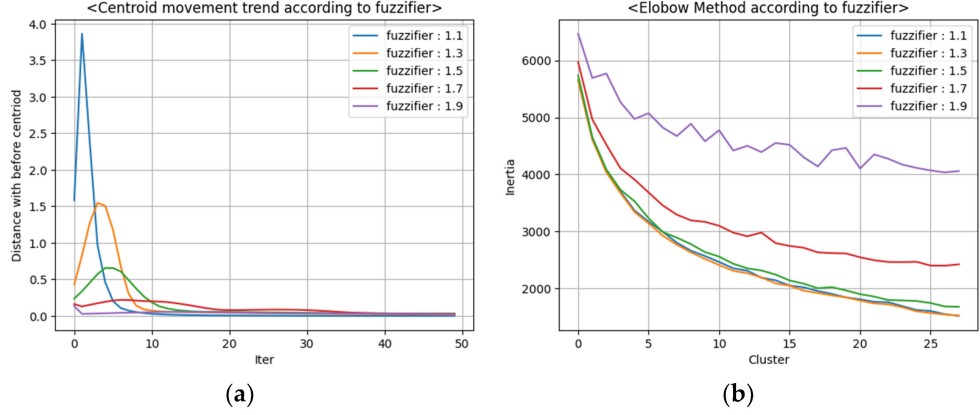

(**a**)                                                    (**b**)

**Figure 10.** (**a**) Result of the centroid movement trend according to fuzzifier and (**b**) result of the inertia trend according to fuzzifier.

The clustering time has a linear relationship with the audience size, with a minimum value at audiences of 1000 and a maximum value at audiences of 10,000. The fuzzifier was set from 1.1 to 1.5 based on the previous experimental results, and the number of clusters was set from 4 to 10, which is the point where the inertia decreases rapidly in Figure 10b. The detailed experimental results are described in Table 3, and an example clustering result can be seen in Figure 11. The experimental results show a linear relationship between crowding and execution time, and the correlation between the execution time and fuzzifier is small. Considering the average network latency of 10–20 ms in Korea, we calculated the maximum latency of 50 ms per zone and, based on audiences of 10,000, we can assume that it takes from 267 ms to 549 ms for audience information to be transmitted and for audience and zone information to be received by the performer in the proposed framework.

**Table 3.** Experiment result of FCM clustering.

| | | Clustering Time (ms) | | | | |
|---|---|---|---|---|---|---|
| | | Fuzzifier | | | | |
| | | 1.1 | 1.2 | 1.3 | 1.4 | 1.5 |
| **Cluster Number** | 4 | 28.01~217 | 24.38~193 | 25~191.56 | 26~189 | 24~193 |
| | 5 | 29~239 | 29~247 | 30~243 | 29~243.1 | 29~244 |
| | 6 | 35~290 | 34~293 | 34.96~311 | 43.31~299 | 35~314 |
| | 7 | 41~364.47 | 40~342.41 | 40~350 | 40~340 | 43~338 |
| | 8 | 49~398 | 47~387 | 46~386 | 47~394 | 49~400 |
| | 9 | 51~444 | 52~445 | 57~437 | 55~438.56 | 52~437 |
| | 10 | 58~502 | 59~492.22 | 64~499 | 57~498 | 57~497 |

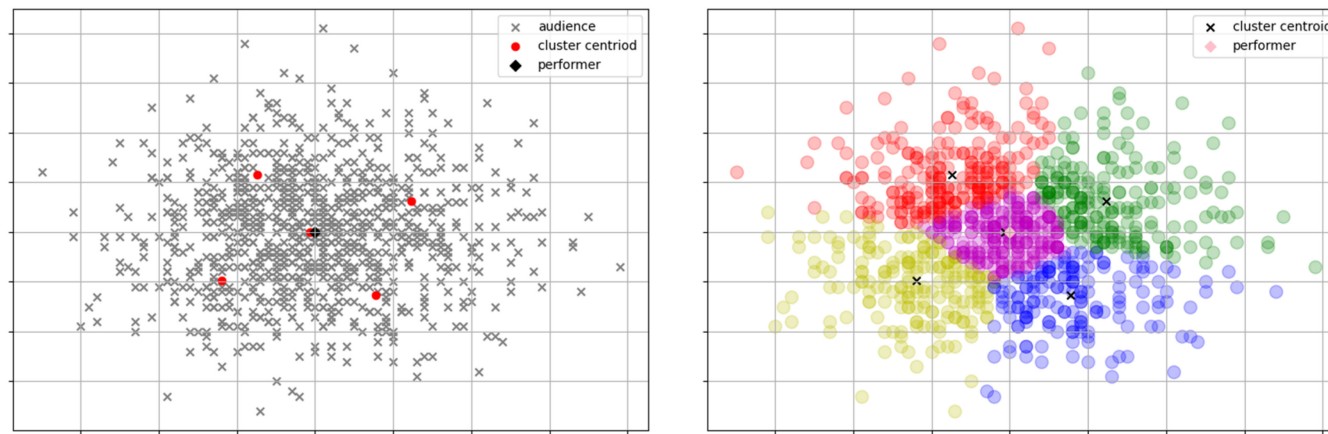

**Figure 11.** Example of the FCM clustering result with clusters represented by different colors.

## 5. Conclusions

This paper proposes a network framework that separates data channels from the existing framework to transmit interaction data between performers and audiences in virtual concerts based on the metaverse platform. In order to provide an immersive experience to the audience, we designed a network using XDN (experience delivery network)-based microservices. Additionally, we presented a task that can visually experience interactions by zone, enabling performers to efficiently receive interactions from a large audience and immerse themselves in virtual concert world. This is done by constructing zones based on the network packet information of the audience and the coordinates in the virtual concert world. In this process, we performed clustering using the fuzzy c-means algorithm to divide the zones according to the varying number of audiences. To reveal the relationship

between the hyperparameters of FCM clustering and the clustering performance by generating virtual audiences, we conducted simulation experiments as in this paper. As a result, we derived a latency of about 267–549 ms for matching the audience information to a zone (cluster) and transmitting it to a performer within the network framework. Furthermore, we established the framework of network design at the SDN (software-defined networking) level and proposed an architecture that can communicate interaction data of virtual concerts in each plane. Additionally, to efficiently transmit interaction data from the performer to a large audience, we designed the network by setting up CDN-based edge servers.

This research is significant in that it precisely designs a network framework dedicated to interactions, which can be incorporated into the network architecture based on the metaverse platform proposed for virtual concerts. Existing studies have operated virtual concerts using game engines that support a single network or metaverse network architectures that provide various services in an integrated manner. However, by establishing a network architecture specialized for the concert industry as a separate category, we approached dealing with large audiences from a network perspective and proposed a solution for it. In order to transmit and receive interaction data between the audience and the performer in a virtual concert, we examined the existing cloud-based network architecture for virtual concerts and designed it in an SDN-based network architecture so that it could be integrated into the system.

As future research, we plan to conduct simulation experiments to enhance the network architecture for transmitting interaction data in a virtual concert architecture configured in a cloud environment. We will assume the total number of audiences for the simulation, configure edge servers in the form of CDN (content delivery network) accordingly, and design a new algorithm capable of handling scalable multi-interaction data by programming the microservices of XDN (experience delivery network). In this process, we will design a task where the performer selects a zone and sends an interaction only to that zone. Furthermore, we will expand the traffic, data volume, and types generated through example virtual concerts and intelligently upgrade the network framework by calculating cases that can occur during the concert.

**Author Contributions:** Conceptualization, C.-R.Y. and I.A.; Project administration, I.-K.J. and T.-W.U.; Supervision, J.K.; Writing—original draft, S.O. and K.C.; Writing—review and editing, J.K. and M.H. All authors have read and agreed to the published version of the manuscript.

**Funding:** This research was supported by the Culture, Sports, and Tourism R&D Program through the Korea Creative Content Agency grant funded by the Ministry of Culture, Sports, and Tourism in 2022 (Project Name: Development of real-time interactive metaverse performance experience platform technology on the scale of a large concert hall Project Number: RS-2022-050002, Contribution Rate: 100%).

**Institutional Review Board Statement:** Not applicable.

**Informed Consent Statement:** Not applicable.

**Data Availability Statement:** Not applicable.

**Conflicts of Interest:** The authors declare no conflict of interest.

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
