# Peer review of "XDN-Based Network Framework Design to Communicate Interaction in Virtual Concerts with Metaverse Platforms"

_applsci, doi:10.3390/app13179509_

Round 1

Reviewer 1 Report

The authors should undertake further study and research on the meaning of Mixed Reality (MR) and Extended Reality (XR). In section 2.3. the authors present the term mixed reality accompanied by the XR acronym, as follows “Services that support mixed reality (XR), such as virtual concerts…”. Also, the authors are advised to briefly explain what mixed reality is, supported by references.

In section 2.3, the authors should further improve the quality of the presented data. In line 149, the authors mention “enables immersive content experiences including XR and metaverse”. The authors should provide examples of these immersive content experiences for the readers to understand the opportunities brought by XDN in the scope of this paper. Also, the authors are advised to gather more references to support the description of XDN. Moreover, the authors should explain what OMAF streaming is, providing a reference.

When presenting Figure 4, the authors should explain it instead of relying solely on the graphics to provide the information. Also, the image should be presented in higher quality, as it is too small and blurry.

Still in section 2.3, line 169, the authors state “CDN architectures can reduce bandwidth consumption and latency, and provide scalability needed for handling abnormal traffic loads, improving server performance”. The authors should include a reference to support this information.

The paper has several structural problems. For instance, before presenting the proposal, the authors start section 2.4, line 174, with “In this paper, we employ clustering algorithms to segment a large audience within a virtual concert based on certain criteria”. This information should be part of the proposal. To avoid this sequentialization issue, the authors can use the introduction paragraph of section 2 to specify why the following subsections 2.1-2.4 relate with their work.

The paper lacks references to support important assumptions presented along the subsections of section 2.

In Table 1, requirements column, the requirement “User motion information (joint variations)” can simply be described as body motion information.

As I understand, Figure 5 is adapted from another work, referenced in the caption. If this was proposed by other authors, it should be included and explained in section 2. Are there any other metaverse network designs to deal with the issue addressed in this paper? Moreover, line 210, the authors briefly analyze a section of Figure 5 without providing a reference to the respective work, where the network design was proposed. Also, the authors should mention why this brief analysis was highlighted. Why is it important to mention that “during the real-time concert, only real-time data provided from the performers are streamed”?

In the middle of the proposal, line 274, the authors write about future work, as in the following sentence “Future research will design a task where the performer selects a zone and sends interactions only to that zone”.

The authors do not provide any kind of evaluation to measure the quality and applicability of their proposal. One of the intentions of this work is to ensure scalability and low latency. However, this is not proven through testing.

In the conclusion, lines 313-315, the authors write “Additionally, we presented a task that can visually experience interactions by zone, enabling performers to efficiently receive interactions from a large audience and immerse themselves in virtual reality”. This sentence includes several problems. First, what does it mean that a task can visually experience interactions? Then, it is the first time that the authors talk about virtual reality. Previously, the authors only wrote about mixed reality and extended reality. It seems that there is confusion regarding the visualization alternatives. Also, the authors did not test their network proposal to state with certainty that it is an efficient approach to receive interactions.

Line 83, the “Defined” in the title is misspelled.

Line 98, Figure 1. Is the figure original from the authors? If adapted from another work, provide the reference explicitly. The same applies to figures 2 and 3.

Line 136, the meaning of the following sentence is not clear: “However, there are limitations with the current CDN structure to facilitate this [13].”

Line 197, the meaning of the following sentence is not clear: “As shown in Table 1, network functions for interactions can be derived.”

Line 199, Table 1, the 3rd word of the caption is misspelled.

Author Response

I sincerely thank you for thoroughly reviewing my paper.

Reviewer 2 Report

This research paper introduces a network framework specifically tailored for virtual concert composition. The primary objective of this framework is to enhance the immersive experience by segregating channels dedicated to interaction, thereby facilitating a dynamic exchange between performers and the audience.

The main weakness of this research lies in the absence of comprehensive validation for the proposed network framework. While this paper primarily focuses on presenting a metaverse network concept, it is imperative to validate the efficacy and feasibility of the proposal through empirical analysis and experimentation. Thus, conducting validation studies becomes a crucial step in further strengthening the credibility and robustness of the proposed framework.

The english writing within this document is moderate, so there exists an opportunity for further refinement.

Author Response

(The authors gave the same response as above.)

Reviewer 3 Report

The authors presented an interesting framework for working in the metaverse. The study provides a clear and concise structure, which makes it easy to follow and understand.

I recommend only some basic editing issues. For example, the second keyword is apparently miss spelled. Also, I recommend reorganizing the keywords in alphabetical order. Finally, the authors should assign an acronym to a term, the first time it is used, and then use only the acronym (the acronym XDN (Experience Delivery Network) was defined multiple times.

The manuscript requires only basic editing regarding English Language.

Author Response

(The authors gave the same response as above.)

Round 2

Reviewer 1 Report

The title should be revised as several words are lower case.
The authors should check if all of the images that are not original include an explicit reference to their source.

Once again, in 2.3, I do not understand what is the meaning of "Services that support mixed reality (XR), such as virtual concerts, provide immersive content to users". Are virtual concerts services that support XR? Or XR provides technologies to support services such as virtual concerts?

Still in the same sentences, the authors fail to fix a previous review. Mixed reality is not XR as written. 
Also, to the review, in comment #1 the authors state "XR is correct to mean augmented reality, not mixed reality". XR is not correct to mean AR.
Therefore, the authors do not show understanding about the visualization alternatives encompassed by XR as an umbrella term.

In comment #2, the authors state "to enable users to immerse themselves in the virtual world environments surrounding them in the metaverse service, the metaverse service developers must provide ultra-high-definition video data of 360-degree views". This is not true. The usage of video is not mandatory in metaverse services.

There seems to be lack of information about metaverse experiences and visualization technologies.

In line 288 of the paper, the authors state "After the concert starts and interaction data is sent from the Node, the process mentioned above is followed until the MIF accumulates interaction data within Zones. A specific time interval is set (generally between 500ms to 3000ms)". 
In their comments, the authors write about being difficult to perform an evaluation of the proposed framework. However, they present time interval values. Where do these values come from? Was a prototype developed?

The english still needs improvement as several sentences seem to mean the opposite of what they are intended to mean.

Author Response

Thank you very much for your comments.
Detailed response can be found in the attached file.

Round 3

Reviewer 1 Report

The author's should add the simulation description as well as the estimated metric values in the paper, so that this work is better supported in terms of feasibility.

The quality of the English seems to have improved a little.

Author Response

Thank you so much for your thoughtful review comment. The review has encouraged us to add more simulations to the paper. Please see the attached response document for more details.
